# Maternal High-Fat Diet Aggravates Allergic Asthma in Offspring via Modulating CD4^+^ T-Cell Differentiation

**DOI:** 10.3390/nu14122508

**Published:** 2022-06-16

**Authors:** Hui Lin, Yiran Zhao, Yajie Zhu, Cheng Li, Wei Xu, Xi Chen, Hefeng Huang, Li Jin

**Affiliations:** 1Obstetrics and Gynecology Hospital, Institute of Reproduction and Development, Fudan University, Shanghai 200011, China; linhui_angela@126.com (H.L.); centrino-li@hotmail.com (C.L.); 2Shanghai Key Laboratory of Embryo Original Diseases, Shanghai 200030, China; zhaoyiran1212@outlook.com (Y.Z.); zhuyajiesjtu@163.com (Y.Z.); 3Research Units of Embryo Original Diseases, Chinese Academy of Medical Sciences, Shanghai 200030, China; 4International Peace Maternity and Child Health Hospital, Shanghai Jiao Tong University School of Medicine, Shanghai 200030, China; 5Department of Immunology, School of Basic Medical Sciences, Fudan University, Shanghai 200032, China; wei_xuxx@fudan.edu.cn; 6Key Laboratory of Reproductive Genetics (Ministry of Education), Zhejiang University, Hangzhou 310006, China

**Keywords:** asthma, CD4^+^ T cells, DNA methylation, fetal origin diseases, maternal high-fat diet

## Abstract

Maternal improper nutrition has been reported to trigger respiratory disorders in offspring. Here, we characterized the effects of high-fat environment in the fetal period on mice and human cord blood CD4^+^ T-lymphocytes, and investigated their roles in susceptibility to asthma. Mice born to mothers that consumed a high-fat diet (HFD) throughout the gestation period were sensitized by ovalbumin to establish an experimental asthma model. To further extrapolate to humans, we collected cord blood from neonates of hypercholesterolemic (HC) mothers (*n* = 18) and control mothers (*n* = 20). In mice, aggravated airway hyperresponsiveness and inflammation revealed that maternal high-fat diet could lead to exacerbated allergic asthma in adult offspring. It was partially due to augmented activation and proliferation of CD4^+^ T-cells, where upregulated *klf2* mRNA levels may be potentially involved. Notably, naïve HFD CD4^+^ T-cells had enhanced T_H_2-based immune response both in vivo and in vitro, resulting from DNA hypomethylation of the *Il-4* promoter region. Moreover, in human, T_H_2 cytokines transcripts were enhanced in CD4^+^ T-cells of the HC group, which was associated with an increased risk of developing allergic diseases at 3 years old. Together, our study indicated that early life improper nutrition-triggered epigenetic changes in T-cells may contribute to long-lasting alterations in allergic diseases.

## 1. Introduction

Epidemiological investigation has demonstrated that 8.4% of the population has asthma according to the National Center for Health Statistics [1]. Causes for increasing respiratory disorders are likely multifactorial, among which the effects of high fat and western food intake on asthma have been well documented [2,3]. Recently, however, much more attention has been paid to the role of maternal diet during pregnancy in asthma development of offspring. The early life environment has the “fetal programming effect” on long-term health [4]. Epidemiological investigations have revealed that high fat intake during pregnancy could result in an alteration of fetal lung development and increase the risk of respiratory disorders in offspring [5,6,7].

Bronchial asthma is featured with chronic airway inflammation and airway hyper responsiveness. The inflammatory phenotype of asthma is characterized by increased production of inflammatory factors, alveolar destruction, elevated mucus secretion, and infiltration of immune cells in the lungs, including eosinophils, type 2 T helper cells (T_H_2), mast cells, and macrophages [8]. T lymphocytes, especially CD4^+^ T-cells and related cytokines have been proved to play an important role in the development of asthma [9,10]. Being sensitized with allergen presented by antigen-presenting cells, naïve CD4^+^ T-cells become activated and differentiate into pro-allergic T_H_2 cells. The cytokines secreted by CD4^+^ T-cells, typically, type 2 cytokines such as IL-4, IL-5, and IL-13, are closely involved in asthma [11]. Animal models and clinical studies, as well as our previous findings have suggested that maternal diets, such as high-fiber diets, vitamin D supplements, and protein-restricted diets influence immune cells in offspring [12,13,14]. However, the influence of maternal high-fat diet on T-lymphocytes in offspring remains unclear.

Epigenetic mechanisms have been reported to be critical in T-cell differentiation and function [15]. Antigen stimulation in naïve CD4^+^ T-cells induces transcription of *Ifng* (T_H_1 cytokine) and the linked *Il-4*, *Il-5*, and *Il-13* genes of the T_H_2 cytokine cluster (*Il-4* locus) [16]. The T_H_ signature cytokine locus is epigenetically activated by DNA demethylation during differentiation. Previous studies have pointed out that early life environment can influence immunity and metabolism system in offspring by altered DNA methylation [13,17]. Thus, we postulate that nutritional environment during pregnancy may influence T-cell differentiation and function in offspring through DNA methylation.

To investigate the influence of the high-fat diet during pregnancy on asthma susceptibility of offspring, both rodent models and cord blood samples of human neonates were used in this work. Pregnant mice were fed a high-fat diet or a normal chow diet throughout the gestation period. Offspring from breeders fed on a normal chow diet or a high-fat diet were defined as NCD or HFD. Adult offspring were then sensitized by ovalbumin (OVA) to establish the experimental asthma model, the severity of which was assessed by airway hyperreactivity and inflammation. To reveal the mechanisms of exacerbated allergic asthma of offspring, immunological characteristics and DNA methylation status of T_H_2 cytokine locus were further analyzed. Meanwhile, to explore the situation in humans, this work analyzed cord blood CD4^+^ T cells and applied follow-up investigation for neonates of hypercholesterolemic (HC) and control (CON) mothers for 3 years.

## 2. Materials and Methods

### 2.1. Mice and Diets

Female C57BL/6 SPF mice, aged 6 to 8 weeks and weighing 18 to 22 g, were obtained from Slack Laboratory Animal Company in Shanghai and raised in SPF animal room (Shanghai Model Organisms Center). Three to five mice were raised in the same cage. Animal experiments were approved by the Pennington Biomedical Research Center (Louisiana State University) and IACUC (approval ID: 2018-0005). Eight-week-old female mice were mated with male mice at a 1:1 ratio. A 60% high-fat diet has previously been used in diet-induced obesity [18]. Dams were supplied with a high-fat diet containing 60% of calories or a matched normal chow diet containing 10% of calories since pregnancy, and switched to the normal chow diet after labor (FBSH Diets Services, Shanghai, China, Appendix A). Offspring were weaned onto a normal chow diet.

### 2.2. OVA-Induced Experimental Asthma

Experimental asthma was induced by OVA. Mice were sensitized using intraperitoneal injection of 50 μg OVA (Grade V; Sigma Aldrich, St. Louis, MO, USA) in 50 μL (2 mg) of alum (Imject Alum; Pierce Biotechnology, Rockford, Ill) on day 0 and day 7, respectively. Then, mice were challenged intranasally with 10 μg OVA (50 μL) dissolved in PBS from day 14 to day 16. Animals in the control group were challenged intranasally with PBS alone. Mice were sacrificed on day 17 for lung airway hyperreactivity measurement and follow-up experiments.

### 2.3. Lung Airway Hyperactivity Measurement

Twenty-four hours after the last challenge, the trachea of adult offspring was exposed under deep anesthesia and an 18-gauge catheter was inserted. The catheter was connected via a nebulizer to a computer-controlled FinePointe RC system (Buxco Electronics, Wilmington, NC, USA). Aerosolized methacholine (Sigma) was nebulized at cumulative doses of 0, 5, 12.5, 25, and 50 mg/mL. Lung resistance was then measured. 

### 2.4. Bronchoalveolar Lavage

Bronchoalveolar lavage fluid (BALF) was first aspirated with 0.5 mL PBS and centrifuged, the supernatant was stored at −80 °C for ELISA. Then, lung lavage was performed by another 2 mL of PBS. Concentrations of IL-4, IL-5, and IL-13 in BALF were measured by ELISA, according to the manufacturer’s instructions (eBioscience, San Diego, CA, USA).

### 2.5. Flow Cytometry

Cells were blocked with FcR antibody (anti-CD16/32, BD Biosciences, San Jose, CA, USA) and stained with conjugated antibodies in PBS containing 2% (wt/vol) BSA. Neutrophils (CD11c^−^SiglecF^−^CD11b^+^Ly6G^+^), eosinophils (CD11c^−^SiglecF^+^), B lymphocytes (CD3^−^CD19^+^), and T lymphocytes (CD3^+^CD19^−^) were detected. For intracellular cytokine analysis, cells were stimulated with phorbol 12-myristate 13-acetate (PMA, 50 ng/mL; Sigma), ionomycin (ION, 500 ng/mL; Sigma) together with GolgiPlug (BD Biosciences) for 5 h. Cells were collected for surface staining of CD4, CD25 (BD Biosciences), the intracellular staining of IL-4, IL-5, IL-13, IFN-γ (eBioscience), and the intranuclear staining of Ki67 (eBioscience). Viability was evaluated with an Annexin-fluorescein isothiocyanate and 7-aminoactinomycin D (7-AAD) Flow kit (BioLegend, San Diego, CA, USA). T-cells were prelabeled with carboxyfluorescein succinimidyl amino ester (CFSE, eBioscience) for detection of proliferation. All flow cytometric data were acquired from BD FACSCelesta™ and analyzed with FlowJo software (TreeStar, San Carlos, CA, USA).

### 2.6. Tissue Histologic Sections

Lung samples from the left upper lobe were fixed with 4% paraformaldehyde, embedded in paraffin, and then sliced into a thickness of 4 microns. Sections were continually stained with hematoxylin and eosin or periodic acid-Schiff or anti-CD4 antibody. 

### 2.7. T-Cell Isolation and Culture

T-cells were obtained from lung, spleen, thymus, or tracheobronchial lymph nodes (tLN). Mouse naïve CD4^+^ T-cells (CD4^+^CD62L^hi^CD44^−^) were sorted by Naïve CD4^+^ T-cell Isolation Kit (Miltenyi Biotec, Bergisch Gladbach, Germany). Human CD4^+^ T-cells were sorted by CD4^+^ T-cell Isolation Kit (Miltenyi Biotec). The purity levels were above 95% detected by BD FACSCelestaTM. For in vitro experiments, mouse CD4^+^ T-cells were cultured in RPMI 1640 medium (Corning, Manassas, VA, USA) supplemented with 10% FBS, and 1% penicillin-streptomycin, and then stimulated by anti-CD3 (precoated, 5 μg/mL; BD Biosciences) and anti-CD28 (1 μg/mL; BD Biosciences) for 3 h. For T_H_2 cell differentiation, naïve CD4^+^ T-cells were cultured under the following conditions: mIL2 (10 ng/mL; BD Bioscience), mIL4 (50 ng/mL; BD Bioscience), and anti-IFN-γ (10 μg/mL; BD Bioscience) with anti-CD3 (precoated, 5 mg/mL) plus anti-CD28 (1 mg/mL; BD Bioscience) for 5–7 days. Human CD4^+^ T-cells were stimulated by PMA (50 ng/mL; Sigma) and ION (500 ng/mL; Sigma) for 3 h. In pharmacologic inhibition experiments, T cells were pretreated with 5-aza-2′-deoxycytidine (1 μg/mL; MedChemExpress, Monmouth Junction, NJ, USA) and then stimulated with anti-CD3 (precoated, 5 μg/mL; BD Biosciences) and anti-CD28 (1 μg/mL; BD Biosciences) for 3 h.

### 2.8. Human Subjects

The study was approved by the International Peace Maternity and Child Health Hospital research ethics committee, the approval number was (GKLW) 2017-81, and the study conforms to the Declaration of Helsinki. Individuals provided their informed consent before their inclusion in the study. Inclusion criteria for hypercholesterolemic mothers (HC) group: plasma cholesterol level greater than or equal to 5.20 mmol/L on 12 gestation weeks. Inclusion criteria for CON group: plasma cholesterol level less than 5.20 mmol/L on 12 gestation weeks. The exclusion criteria of HC and CON mothers: asthma, atopic dermatitis, allergy to food, gestational diabetes, gestational hypertension diseases, preterm birth or overdue pregnancy, fetal abnormalities, body mass index (BMI) > 24.9 kg/m^2^ or <18.5 kg/m^2^ before pregnancy, and polycystic ovary syndrome. Umbilical cord blood samples from the HC (*n* = 18) and CON (*n* = 20) groups were collected in 2017. Cholesterol, triglyceride, and low-density lipoprotein concentration in maternal plasma were detected on 12 gestation weeks. Parental reports of physician-diagnosed allergy diseases, including asthma or recurrent wheezing, atopic dermatitis, and food allergy were collected via a telephone questionnaire at the 3-year follow-up. 

### 2.9. Relative mRNA Expression Analysis

Quantitative PCR was performed, as previously described [19], and primers for *Il-4* (ID 16189), *Il-5* (ID 16191), *Il-10* (ID 16153), *Il-13* (ID 16163), *Ifng* (ID 15978), *klf2* (ID 16598), *p53* (ID 22059), *hif1α* (ID 15251), *cyclind2* (ID 12444), and *cyclind3* (ID 12445) in mouse and *Il-4* (ID 3565), *Il-5* (ID 3567), *Il-13* (ID 3596), and *Ifng* (ID 3458) in human (see Appendix A) were from Primer Bank. Data were normalized to, respectively, mouse *Rpl13a* (ID 22121) or human *β actin* (ID 60), and results were shown as fold induction relative to expression levels in PBS-treated or naïve tissues, as previously indicated. Gene IDs were from GenBank.

### 2.10. Bisulfite Sequencing

The CpG DNA methylation status of the mouse *Il-4* promoters was measured as previously described [13]. The methylation status of the human *Il-4* promoters was assessed by a next-generation sequencing-based BSP, according to a previously published method [20]. The primers used are shown in Appendix A.

### 2.11. Statistical Analysis

A one-way analysis of variance (ANOVA) or a two-tailed Student’s *t*-test was used to analyze the experimental results between groups. Data were summarized as mean ± SEM. Categorical variables are expressed as a frequency with a proportion, and the differences in these were detected by Fisher’s exact test. Significance was defined as *p* < 0.05. All data are shown as mean ± standard error.

## 3. Results

### 3.1. Maternal High-Fat Diet during Pregnancy Aggravated Experimental Asthma in Offspring

We used an OVA-induced experimental asthma model in this study. Offspring from breeders fed on a normal chow diet or a high-fat diet were defined as NCD or HFD throughout the study. NCD and HFD mice displayed comparable baseline airway resistance (R_L_), while challenging with 25 mg/mL or 50 mg/mL aerosolized methacholine led to a significantly enhanced level of R_L_ in OVA-challenged HFD (HFD-OVA) mice (Figure 1A). The immune cells in respiratory tissues were then analyzed. Increased total infiltrated leukocytes, especially eosinophils, were found both in BALFs and lungs in HFD-OVA mice (Figure 1B,C). Consistently, HFD-OVA mice displayed exacerbated tissue structure destruction and increased mucin production in the airway epithelium (Figure 1D,E). These results collectively indicated that maternal high-fat diet during pregnancy aggravated OVA-induced allergic asthma in offspring. 

### 3.2. Maternal High-Fat Diet during Pregnancy Promoted CD4^+^ T-Cell Proliferation and Activation in Offspring

T-cells play a key role in the process of allergic asthma [21,22]. We first analyzed the total T-cells and major subsets in thymus and periphery. The numbers of CD4^+^ and CD8^+^ T-cells were comparable in thymus and spleen (Appendix A). Furthermore, subset analysis of splenic CD4^+^ and CD8^+^ T-cells showed no difference in naïve or memory T-cell homeostasis (Appendix A). T_reg_ cells, the main negative regulatory cells of the immune response [23], were comparable in HFD and NCD (Appendix A).

We then analyzed lymphocytes in vivo by OVA model. We found that HFD mice showed significantly increased T-cells infiltration, especially CD4^+^ T-cell infiltration in both lungs and tLNs (Figure 2A–C). Increased proliferation was then found in HFD CD4^+^ T-cells by highly expressed proliferation marker Ki67 (Figure 2D–F), while viability was comparable between the two groups measured by Annexin V and 7-AAD (Figure 2G,H). 

We next examined TCR-induced activation and proliferation in naïve CD4^+^ T-cell (CD4^+^CD62L^high^CD44^−^; unless otherwise noted, mice naïve T-cells were used throughout the study) in vitro. Notably, increased proliferation was observed in HFD CD4^+^ T-cells as measured by cell enumeration (Figure 2I) and by tracking CFSE dilution (Figure 2J). In the meantime, increased activation was also found by highly expressed activation marker CD25 (Figure 2K). In contrast, apoptosis and cell death remained unchanged (Figure 2L,M). We next analyzed genes involved in modulating proliferation of T-cells described by previously published studies [24,25,26]. Kruppel-like factor 2 (KLF2) was regulated by multiple microenvironments including low-density lipoprotein and played an important role in proliferation and differentiation in immune cells [27]. We found that the mRNA expression level of *klf2* was significantly upregulated in HFD CD4^+^ T-cells. The mRNA expression levels of *Cyclind2* and *Cyclind3* were also elevated (Figure 2N). Taken together, our data suggested that maternal high-fat diet during pregnancy enhanced HFD CD4^+^ T-cells activation and proliferation in offspring

### 3.3. Offspring of Maternal High-Fat Diet during Pregnancy Exhibited Increased T_H_2 Cytokine Level

The balance between T_H_1 and T_H_2 is important in the development of asthma. We found that mRNA expression level and BALF concentration of T_H_2 cytokines, including IL-4, IL-5, and IL-13, was significantly increased in HFD-OVA mice, while the expression of T_H_1 cytokine was comparable to NCD-OVA (Figure 3A,B). After secondary stimulation ex vivo, we observed increased IL-4 and IL-13 production by CD4^+^ T-cells from lung-draining lymph nodes and tLNs of HFD-OVA mice, while the level of IL-5 as well as IFN-γ were approximate to NCD-OVA (Figure 3C–F). In vitro, purified naïve CD4^+^ T were magnetically sorted out from spleen and stimulated by anti-CD3/anti-CD28. HFD exhibited increased T_H_2 cytokine profile transcripts (Figure 3G). When cultured under T_H_2 differentiation condition, naïve HFD CD4^+^ T-cells showed enhanced proliferation and increased T_H_2 cytokines transcripts (Figure 3H–K), indicating T_H_2-biased differentiation. Taken together, these results suggested that HFD naïve CD4^+^ T-cells were prone to differentiate into T_H_2 cells.

### 3.4. Maternal High-Fat Diet during Pregnancy Resulted in Hypomethylation of Il-4 Promoter Region

Using the bisulfite sequencing technique, we observed that several CpG islands at the *Il-4* promoter region in HFD CD4^+^ T-cells were relatively hypomethylated (Figure 4A–D). However, the DNA methylation status of CpG islands in the conserved noncoding DNA sequence 1 (CNS 1), *Il-4* intron 1 and exon 2 locus were found to be comparable between NCD and HFD CD4^+^ T-cells (Appendix A). The IL-4 and IL-13 genes reside only 13 kb apart and coordinate regulation of them determines the typical immune responses observed during infection [15]. To further confirm our conclusion, we treated NCD and HFD CD4^+^ T-cells with the DNA-demethylating agent 5-aza-2′-deoxycytidine. We found the *Il-4* and *Il-13* mRNA expression levels in NCD CD4^+^ T-cells were elevated to a comparable level of that in HFD CD4^+^ T-cells. In contrast, in demethylating agent-treated HFD CD4^+^ T-cells, the *Il-4* and *Il-13* mRNA expression levels were approximate to the untreated HFD CD4^+^ T-cells (Figure 4E). Collectively, our results indicated that increased T_H_2 cytokine expression of CD4^+^ T-cells in HFD mice might result from pre-existing hypomethylation in the *Il-4* promoter region.

### 3.5. Neonates of Hypercholesterolemic Mothers Expressed Elevated Levels of T_H_2 Cytokines and were Prone to Allergic Diseases

To explore the situation in humans, we recruited 38 mother–child pairs (HC *n* = 18 and CON *n* = 20) for further study. A comparison of maternal and neonatal variables was listed in Appendix A. We found that triglyceride and low-density lipoprotein were significantly higher in HC mothers (Figure 5A) at 12 gestational weeks. Transcripts of T_H_2 cytokines was markedly increased in cord blood CD4^+^ T-cells in the HC group (Figure 5B). Allergic diseases beginning in adults might have originated in childhood [28]. Follow-up investigation showed that neonates in HC group were prone to develop allergic diseases in childhood (Figure 5C and Table 1), although the differences did not reach statistical significance (*p* = 0.09). Of all the participants, one neonate in the HC group was diagnosed with both asthma and atopic dermatitis, and was surprisingly found to have the highest expression level of *Il-4* in cord blood CD4^+^ T-cells. Thus, we further analyzed the relationship between T_H_2 cytokine mRNA expression and allergic diseases later in life. Subgroup analyses showed that higher *Il-4* and *Il-13* transcripts in cord blood CD4^+^ T-cells were associated with an increased risk of allergic diseases in children of the HC group (Figure 5D). Collectively, our results suggested a correlation between intrauterine improper nutrition environment and susceptibility to allergic diseases in offspring later in life via enhanced CD4^+^ T-cells function.

## 4. Discussion

Maternal diet during pregnancy has a long-term influence on offspring. This study highlights the importance of maternal high-fat diet during pregnancy in increasing susceptibility to allergic asthma in offspring, characterized by increased AHR and airway inflammation. The key finding was that maternal high-fat diet during pregnancy enhanced CD4^+^ T-cell activation, proliferation and T_H_2 skewing in offspring, thus promoting asthma development. Mechanistically, DNA hypomethylation was found in the *Il-4* promotor region in naïve HFD CD4^+^ T-cells, suggesting that enhanced T_H_2 function is an intrinsic property. Consistent with observations in the mouse model, human cord blood CD4^+^ T-cells from hypercholesterolemic mothers showed enhanced mRNA expression of T_H_2 cytokines, which was associated with an increased risk of developing allergic diseases later in life.

Several studies have suggested the role of in utero nutrition in asthma susceptibility of offspring. Thorburn et al. found that a maternal high-fiber diet protected against adult asthma [12]. Our previous findings showed that how maternal undernutrition remodeled cell-intrinsic metabolic pathways within T-cells of offspring [13]. Recently, MacDonald et al. showed similar findings that maternal pre-pregnant obesity led to bronchial hyperresponsiveness in offspring [29]. However, the present study provided stronger evidence of the “developmental origin” of asthma, namely, HFD exposure during pregnancy alone was sufficient to predispose offspring to allergic asthma development later in life. 

To our knowledge, our study demonstrated for the first time that a maternal high-fat diet during pregnancy reprogramed CD4^+^ T-cells to trigger asthma. As proved by previous studies, an imbalanced ratio of T_H_2/T_H_1 significantly affected the occurrence of allergic diseases [30]. We found that maternal high-fat diet promoted T-cell activation, proliferation, and rendered naïve CD4^+^ T-cells prone to T_H_2 differentiation. The surge of type 2 cytokine promoted the infiltration of inflammatory cells into the airway [11]. In return, infiltrated inflammatory cells generated more cytokines, and thus airway responses were aggravated and other allergic reactions were further induced [31]. Previously, the adverse pregnant environment had been reported to influence T_H_2 functioning. Singh et al. found that prenatal secondhand cigarette smoke promoted T_H_2 polarization by activating pathways of GATA3/Lck/ ERK1/2/STAT6, and suppressed the T_H_1 polarization by inhibiting T-bet pathway [32]. Yue et al. found that maternal exposure to NO_2_ caused a striking increase in the lung inflammatory cell infiltration and the release of type 2 cytokines in offspring [17]. Our results further tested the hypothesis that the early stage of life is a key period of immune system development, where the intrauterine nutrition environment affected asthma susceptibility later in life via altered T-cells differentiation and function. 

Intrauterine nutrition can have long effects on adult phenotypes. We demonstrated that the *Il-4* promoter region was at hypomethylation in HFD CD4^+^ T cells, and the difference could be eliminated by a DNA-demethylating agent. Therefore, it can be speculated that hypomethylation induced by intrauterine high-fat may in part be involved in the T_H_2-biased differentiation in mice.

We also provided evidence that our findings on the developmental origin of asthma could be extrapolated and applied to humans. Since it is against medical principles to provide whole-course high-fat food for pregnant women, and considering maternal high-fat diet always resulted in high cholesterol in serum, our project included hypercholesterolemic mothers and their neonates as study objects. We observed enhanced T_H_2 cytokines expression in cord blood CD4^+^ T-cells in human HC group. The follow-up investigation revealed that those who had exuberated T_H_2 cytokine expression were more susceptible to allergic diseases. However, the analysis in the *Il-4* promoter region of human cord blood CD4^+^ T-cells did not show any difference between HC and CON groups (Appendix A), partially due to the overall hypermethylated status in the immune response-related genes in fetal cells [33], where the incidence of methylation clones in most of the detected CpG sites was higher than 95%. Although the sample size was relatively small, these results indicated that cord blood T_H_2 cytokine expression may become a potential predictor for future allergic diseases.

In summary, our findings showed that maternal high-fat diet during pregnancy promoted the proliferation and effector function of T_H_2-cells via the hypomethylation of *Il-4* promoter region to trigger allergic asthma in offspring. These findings explained one aspect of the “fetal origin” of asthma, via diet and epigenetic effects. Although further studies are needed to decipher the molecular links between in utero nutrients and asthma, our study suggested the important role of early-stage nutrition in modulating CD4^+^ T-cell functions in both murine models and human beings. These results provide evidence for dietary instruction during pregnancy and offer new insights into the pathogenesis of developmental origins of adult immune diseases.

## Figures and Tables

**Figure 1 nutrients-14-02508-f001:**
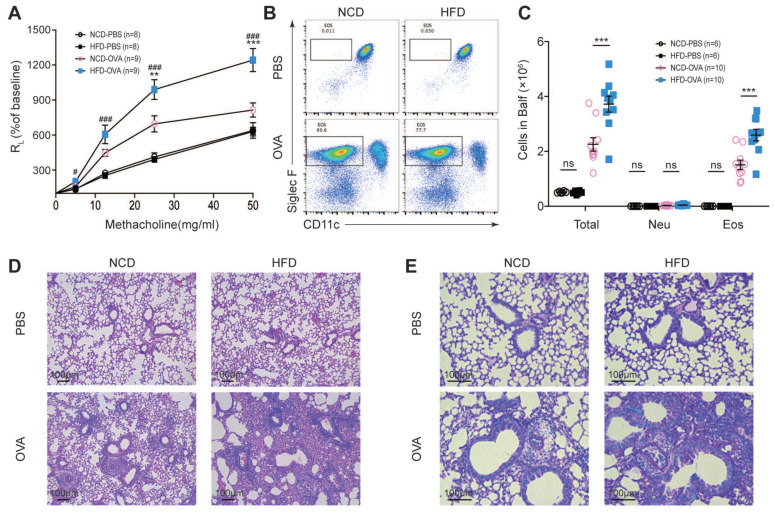
Maternal high-fat diet during pregnancy aggravated experimental asthma in offspring. Mice were immunized intraperitoneally and then challenged intranasally with ovalbumin (OVA) or PBS. Mice were sacrificed 24 h after the last challenge (OVA model). Offspring from breeders fed on a normal chow diet or a high-fat diet were defined as NCD or HFD throughout the study. (**A**) Airway hyperresponsiveness in terms of airway resistance (R_L_), ** *p* < 0.01, *** *p* < 0.001, NCD-OVA versus HFD-OVA, and # *p* < 0.05, ### *p* < 0.001, HFD-PBS versus HFD-OVA. (**B**) Percentages of eosinophils and (**C**) quantifications of immunocytes in Bronchoalveolar lavage fluids (BALFs). Microscopy of lung sections stained with H&E (**D**) and with PAS (**E**). Scale bars represent 100 μm. *p* values were determined by a one-way ANOVA with Tukey multiple comparisons *p* value correction (**A**,**C**). Each dot denotes a value acquired from a single mouse. Data are shown as means ± SEM. *n*, Number of mice in each group. *** *p* < 0.001. *ns*, not significant.

**Figure 2 nutrients-14-02508-f002:**
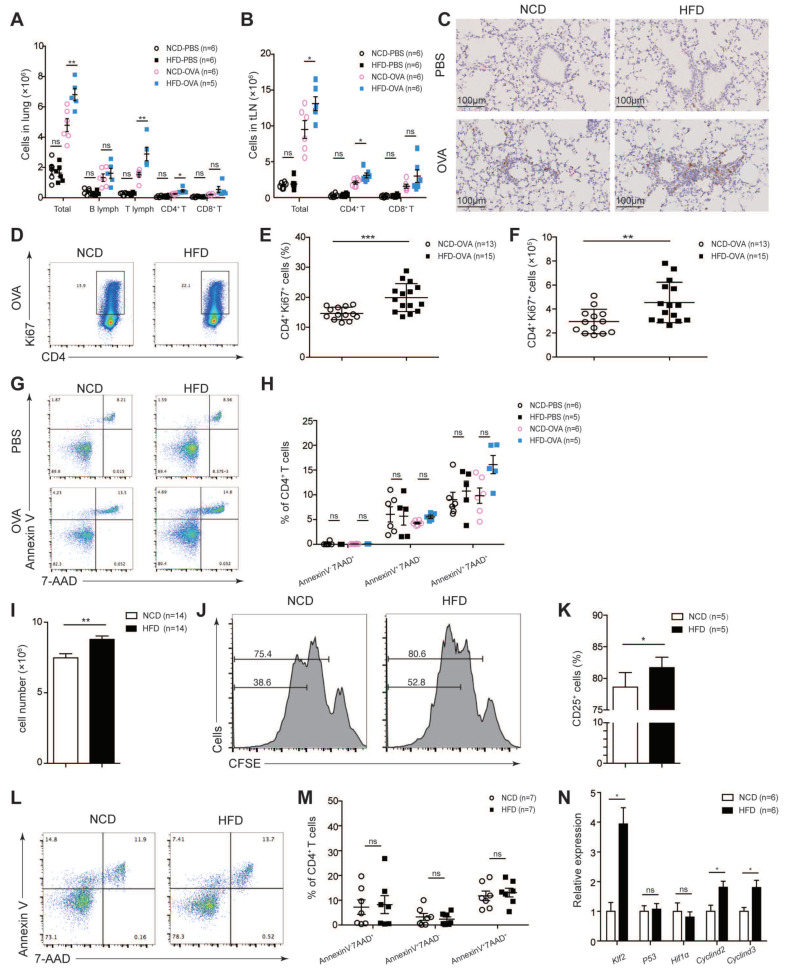
Maternal high-fat diet during pregnancy promoted CD4^+^ T-cell proliferation and activation in offspring. (**A**–**H**) OVA model. Quantifications of immunocytes in lungs (**A**), tracheobronchial lymph nodes (tLNs) (**B**), and microscopy of lung sections stained with CD4 antibody (**C**) were analyzed. Scale bars represent 100 μm. Percentages (**D**,**E**) and quantifications (**F**) of Ki67^+^CD4^+^ T-cells, the Annexin V and 7-aminoactinomycin D (7-AAD) staining of CD4^+^ T-cells (**G**,**H**) were measured in tLNs. (**I**–**N**) Naïve CD4^+^ T-cells were sorted out from spleen and stimulated with anti-CD3/anti-CD28 in vitro. T-cell numbers (**I**), carboxyfluorescein succinimidyl amino ester (CFSE)-labeled T-cells (**J**), the T-cell activation marker CD25 (**K**), the Annexin V and 7-AAD staining of T-cells (**L**,**M**), and relative mRNA expression level of proliferation related gene (**N**) were analyzed on day 3. *p* values were determined through a one-way ANOVA with Tukey multiple comparisons *p* value correction (**A**,**B**,**H**) or a Student’s *t*-test (**E**,**F**,**I**,**K**,**M**,**N**). Each dot denotes a value acquired from a single mouse. Data are shown as means ± SEM. *n*, Number of mice in each group. * *p* < 0.05, ** *p* < 0.01 and *** *p* < 0.001. *ns*, not significant.

**Figure 3 nutrients-14-02508-f003:**
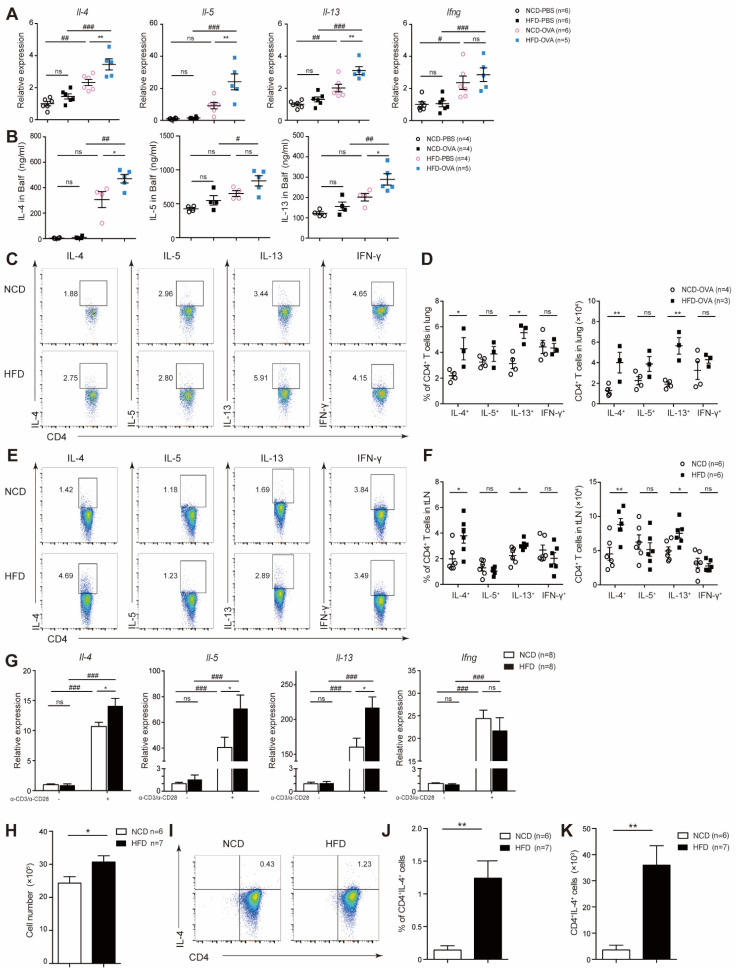
Offspring of mice fed high-fat diet during pregnancy exhibited increased T_H_2 cytokine level. (**A**–**F**) OVA model. (**A**) *Il-4*, *Il-5*, and *Il-13* mRNA expression levels in lungs. (**B**) Concentration of IL-4, IL-5, and IL-13 in BALFs. CD4^+^ IL4^+^ T-cells, CD4^+^ IL5^+^ T-cells, CD4^+^ IL13^+^ T-cells, and CD4^+^ IFNγ^+^ T-cells in lungs (**C**,**D**) and tLNs (**E**,**F**). (**G**–**K**) Naïve CD4^+^ T-cells were sorted out from spleen and stimulated with anti-CD3/anti-CD28 in vitro. (**G**) *Il-4*, *Il-5*, *Il-13*, and *Ifng* mRNA expression levels on stimulation for 3 h. (**H**–**K**) Splenic naïve CD4^+^ T-cells were differentiated under type 2 T helper cells (T_H_2)-skewed conditions for 5–7 days. (**H**) Cell numbers on day 6. Cells were collected and restimulated by phorbol 12-myristate 13-acetate (PMA), ionomycin, and GolgiPlug for 5 h. Percentage (**I**,**J**) and quantifications (**K**) of CD4^+^IL4^+^ T-cells. *p* values were determined by a one-way ANOVA with Tukey multiple comparisons *p* values correction (**A**,**B**,**G**) or a Student’s *t*-test (**D**,**F**,**H**,**J**,**K**). Each dot denotes a value acquired from a single mouse. Data are shown as means ± SEM. *n*, Number of mice in each group. * *p* < 0.05, ** *p* < 0.01, NCD versus HFD. # *p* < 0.05, ## *p* < 0.01, ### *p* < 0.001, OVA versus PBS. *ns*, not significant.

**Figure 4 nutrients-14-02508-f004:**
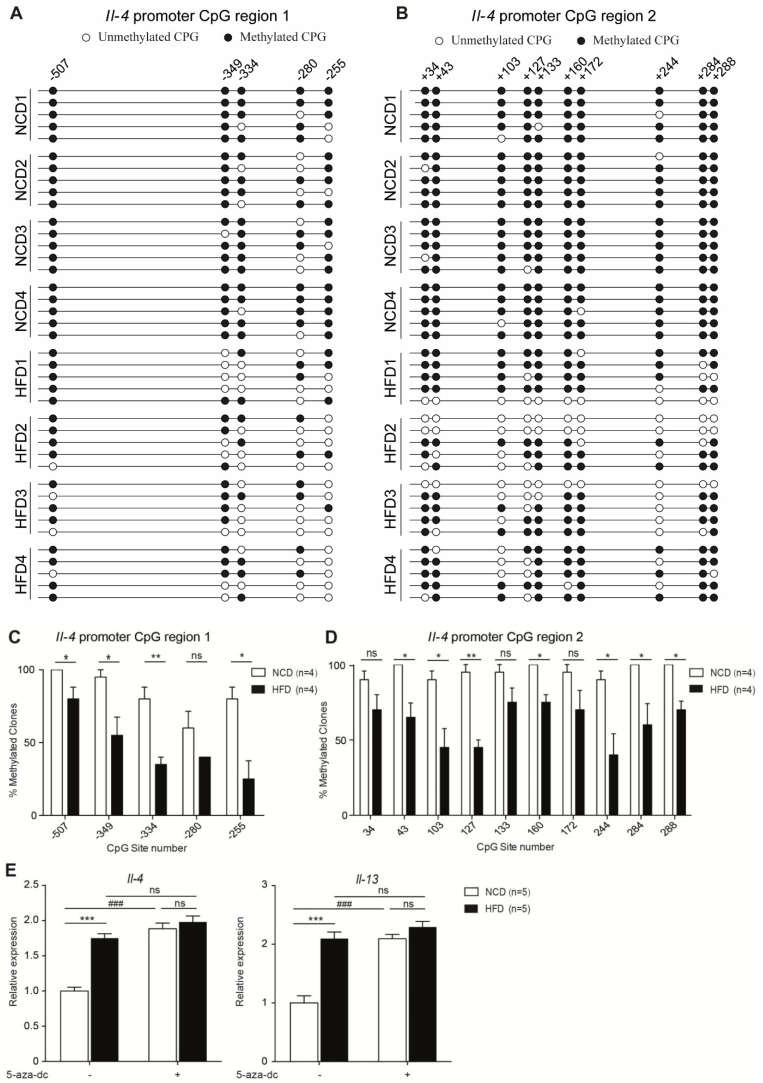
Maternal high-fat diet during pregnancy resulted in hypomethylation of *Il-4* promoter region. Naïve CD4^+^ T-cells were assessed for *Il-4* promoter region methylation level via bisulfite sequencing (**A**–**D**). Representative data and quantifications of methylation at CpG sites of the *Il-4* promoter region 1 (**A**,**C**) and *Il-4* promoter region 2 (**B**,**D**) in naïve CD4^+^ T-cells are shown. CD4^+^ T cells were stimulated with anti-CD3/anti-CD28 for 3 h in the presence of 5-aza-2′-deoxycytidine. The levels of *Il-4* and *Il-13* mRNA expression were analyzed by Real-time PCR analysis (**E**). *p* values were determined by using Student’s *t*-test (**A**–**D**) or one-way ANOVA with Tukey multiple comparisons *p* value correction (**E**). Data are shown as means ± SEM. *n*, Number of mice in each group. * *p* < 0.05, ** *p* < 0.01, *** *p* < 0.001, HFD versus NCD. ### *p* < 0.001, 5-aza-2′-deoxycytidine versus PBS. *ns*, not significant.

**Figure 5 nutrients-14-02508-f005:**
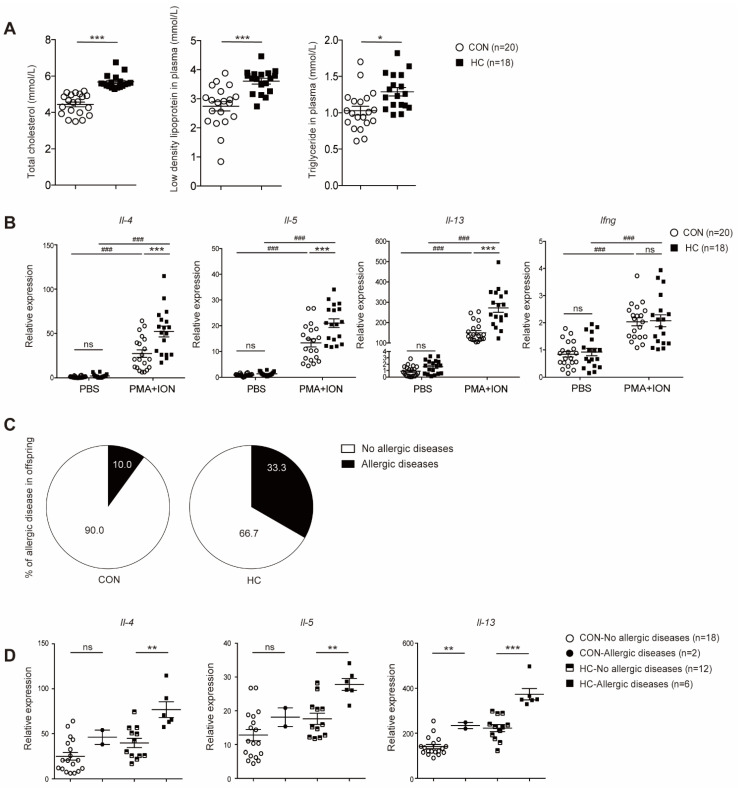
Neonates of hypercholesterolemic mothers expressed elevated levels of T_H_2 cytokines and were prone to allergic diseases. Umbilical cord blood samples from neonates of hypercholesterolemic mothers (HC, *n* = 18) and control mothers (CON, *n* = 20) were collected. CD4^+^ T-cells were obtained by magnetic sorting. (**A**) Total cholesterol, triglyceride, and low-density lipoprotein concentration in maternal plasma on 12 gestational weeks. (**B**) *Il-4*, *Il-5*, *Il-13*, and *Ifng* mRNA expression levels in cord blood CD4^+^ T-cells stimulated with PMA (50 ng/mL) and ionomycin (500 ng/mL) for 3 h. Participants were followed up for 3 years (**C**,**D**). (**C**) Percentage of parental reports of physician-diagnosed allergic diseases in offspring. (**D**) Relation of T_H_2 cytokine mRNA expression levels in stimulated cord blood CD4^+^ T-cells and onset of allergic diseases in childhood. *p* values were determined by a one-way ANOVA with Tukey multiple comparisons *p* value correction (**B**) or a Student’s *t*-test (**A**,**D**). Data are shown as means ± SEM. *n*, Number of participants in each group. * *p* < 0.05, ** *p* < 0.01, *** *p* < 0.001, CON versus HC. ### *p* < 0.001, PMA and ionomycin versus PBS. *ns*, not significant.

**Table 1 nutrients-14-02508-t001:** Incidence of allergic diseases.

	CON *n* (%)	HC *n* (%)	*p* Value
Allergy diseases			0.09
No	18(0.90)	12(66.67)	
Yes	2(0.10)	6(33.33)	
Asthma			0.47
No	20(100)	17(94.44)	
Yes	0	1(5.56)	
Atopic dermatitis			0.40
No	18(0.90)	14(77.78)	
Yes	2(0.10)	4(22.22)	
Food allergy			0.17
No	19(0.95)	14(77.78)	
Yes	1(0.05)	4(22.22)	

Note: Neonates were followed up for 3 years. The primary outcome was a parental report of physician-diagnosed allergic diseases. Categorical variables are expressed as a frequency with a proportion. *p* values were determined by Fisher’s exact test.

## Data Availability

All data relevant to the study are included in the article.

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
