# Peer review of "Maternal High-Fat Diet Aggravates Allergic Asthma in Offspring via Modulating CD4+ T-Cell Differentiation"

_nutrients, 2022, doi:10.3390/nu14122508_

Round 1

Reviewer 1 Report

The manuscript is well written, interesting and reasoning the hypothesis  on multiple stages of scientific investigation.

Minor points:

Abstract requires restructuring. The mouse and human part should be mentioned in first part  - responding to material and methods, and the results from both models should be given as follows.

It is misleading to use the rem malnutrition in regard to high fat diet, I suggest “improper nutrition” or “incorrect diet”

Also the term “provokes”” seems toe out of place, better use “trigger”

Introduction

Line 34 asthma accounted for 8,4% of population – it seems to better  use -

“8,4% of population is accounted for asthma”

The last part of  the introduction line 64-76 – only aim and hypothesis (what was studied in which group) should be given here, not the results – the part from lines 67 – 76 should be removed.

Material and methods

Line 81 Animal studies was proved.  Shouldn’t it be “consented” or “approved”?

Please change “cord blood” subheading to human subjects.

Real time analysis – since it is relative expression (mRNA expression) it seems  that term will be better subheading here.

Results

Lines 271 – 272 It’s not clear why this conclusion was drawn. This requires some explanation in discussion.

Discussion

Line 339 I think it is rather “to trigger” asthma not aggravate.

Line 354 -358 It’s not clear whether authors are speaking about germline effect or intrauterine effect – those are different exposures, still both present in one subject.

Author Response

Dear Reviewer,

Thank you for giving us the opportunity to revise our manuscript (nutrients-1665750) “Maternal high-fat diet aggravates allergic asthma in offspring via modulating CD4+ T-cell differentiation”. We gratefully appreciate your time and efforts. The comments are all of the great value and indeed helpful for improving our paper and providing guidance for further research. Based on the instructions provided in the email, we uploaded the file of the revised manuscript with all the changes tracked. Please see the attachment.

We indeed appreciate your valuable comments and suggestions and hope our responses could satisfy the reviewers and editors to meet the quality standard of Nutrients. Thank you again for your consideration and we are looking forward to your decision.

Reviewer 2 Report

Hui Lin et al have carried out a study of great interest in which they analyse in depth that Maternal high-fat diet aggravates allergic asthma in offspring via modulating CD4+ T-cell differentiation. However, some aspects must be reviewed to enhance the quality of your study.

1. In line 245, the description of their results is very general in terms of the production of type 2 cytokines, so the differences between them should be better explained. Referring to the differences between the production of IL-4 and IL-13 by the T-cells from lung-draining lymph nodes and tLNs of HFD-OVA mice (Fig 3, C-F).

2. Have authors studied whether these proliferative CD4+T-cells areTh2- or TH2A-cells? It would be interesting if the production of IL-4, IL-13…is produced by them.

3. An interesting cell type to study in this study would be ILC2. These provide a potent source of type 2 effector cytokines in the initiation of immune responses. In fact, these cells are in the focus of attention in the pathogenesis of the allergic asthma. The authors have the possibility to carry out some measurements?. For example, analyzing their frequency in HC vs CON.

4. To enhance the authors' results, where they conclude that Treg cells do not show changes between NCD vs HFD. Has it been possible to measure the levels of IL-10 as cytokine of a regulatory pattern?

5. What happens to IgE levels?

Author Response

(The authors gave the same response as above.)

Round 2

Reviewer 2 Report

The study of Hui Lin et al, has been improved, and the replies regarding my comments are explained and I agree with them.